# Systematic Review: Drug Repositioning for Congenital Disorders of Glycosylation (CDG)

**DOI:** 10.3390/ijms23158725

**Published:** 2022-08-05

**Authors:** Sandra Brasil, Mariateresa Allocca, Salvador C. M. Magrinho, Inês Santos, Madalena Raposo, Rita Francisco, Carlota Pascoal, Tiago Martins, Paula A. Videira, Florbela Pereira, Giuseppina Andreotti, Jaak Jaeken, Kristin A. Kantautas, Ethan O. Perlstein, Vanessa dos Reis Ferreira

**Affiliations:** 1UCIBIO—Applied Molecular Biosciences Unit, School of Science and Technology, NOVA University of Lisbon, 2819-516 Caparica, Portugal; 2Associate Laboratory i4HB—Institute for Health and Bioeconomy, School of Science and Technology, Nova University of Lisbon, 2829-516 Caparica, Portugal; 3CDG & Allies PPAIN—Professionals and Patient Associations International Network, Department of Life Sciences, School of Science and Technology, NOVA University of Lisbon, 2819-516 Caparica, Portugal; 4Institute of Biomolecular Chemistry, National Research Council of Italy, 80078 Pozzuoli, Italy; 5Department of Environmental, Biological and Pharmaceutical Sciences and Technologies, University of Campania “Luigi Vanvitelli”, 81100 Caserta, Italy; 6LAQV-Requimte, Chemistry Department, School of Science and Technology, Nova University of Lisbon, 2819-516 Caparica, Portugal; 7Sci and Volunteer Program from School of Science and Technology, NOVA University of Lisbon, 2819-516 Caparica, Portugal; 8Center for Metabolic Diseases, Department of Pediatrics, KU Leuven, 3000 Leuven, Belgium; 9Sappani Foundation, Brampton, ON L6Y 5T2, Canada; 10Perlara PBC, Berkeley, CA 94705, USA

**Keywords:** congenital disorders of glycosylation, drug repositioning, AI in drug discovery, orphan drugs, disease models, biomarkers

## Abstract

Advances in research have boosted therapy development for congenital disorders of glycosylation (CDG), a group of rare genetic disorders affecting protein and lipid glycosylation and glycosylphosphatidylinositol anchor biosynthesis. The (re)use of known drugs for novel medical purposes, known as drug repositioning, is growing for both common and rare disorders. The latest innovation concerns the rational search for repositioned molecules which also benefits from artificial intelligence (AI). Compared to traditional methods, drug repositioning accelerates the overall drug discovery process while saving costs. This is particularly valuable for rare diseases. AI tools have proven their worth in diagnosis, in disease classification and characterization, and ultimately in therapy discovery in rare diseases. The availability of biomarkers and reliable disease models is critical for research and development of new drugs, especially for rare and heterogeneous diseases such as CDG. This work reviews the literature related to repositioned drugs for CDG, discovered by serendipity or through a systemic approach. Recent advances in biomarkers and disease models are also outlined as well as stakeholders’ views on AI for therapy discovery in CDG.

## 1. Introduction

Drug repositioning (also called drug repurposing, reprofiling, or re-tasking) involves the use of approved or investigational drugs for a different application than the original one. This drug development strategy is based on the potential of each compound to interact with distinct targets. Thus, a certain drug can be reused on the same target that is involved in another disease, or on a different target from the primary one. Even the off-target effect of a pharmaceutical product can be harnessed for another indication [1,2].

The standard drug development process can take up to 15 years. Drug repositioning can accelerate this time-consuming process by approximately five years. The drug repositioning strategy decreases the risk and increases the success rate, which on average is approximately 2% in the standard development process. It also reduces the price of drug development and production from approximately $12 billion for the cost of the traditional strategy to $1.6 billion for the repositioning strategy, which is crucial in the rare disease scenario [3].

Many rare diseases (RDs), affecting a small percentage of the population, are multi-system and life threatening. A common definition is still lacking: a disease is considered rare in the United States with a prevalence of less than 0.08%, in Japan less than 0.04% and in Europe less than 0.05% [4,5]. More than 10,000 diseases are recognized as rare which, although individually rare, affect about 400 million people worldwide [6]. Despite global efforts [7,8], it is estimated that less than 5% of RDs have an approved therapy, mainly due to the low number of people affected, the heterogeneity of the disease presentation, and the lack of scientific data supporting a pharmaceutical strategy [9].

Drug repositioning mostly arises from accidental observations or informed insights from previous studies. Although it has become popular in recent years, it is still restricted to therapeutic areas making greater economic profits, such as cancer and other more prevalent diseases [2]. A rational design of repurposed molecules could help to extend drug repositioning to a variety of diseases [10]. In addition, continuous computational advances offer many opportunities for a systematic search for repurposed therapeutics. For example, many compounds can be virtually tested simultaneously, investigating new potential targets and their associated pathways [9].

The development of therapies for RDs may greatly benefit from allying drug repositioning with artificial intelligence (AI) technology [11]. AI is an ever-growing field and its application in biomedicine has been expanding in recent years [12]. It enhances drug development, making it more efficient and less costly [13]. Some recent reviews on AI applied to RD drug discovery give insights in current medical applications, types of algorithms and input data as well as which diseases are being studied [11,14]. The increase in the available information derived from new technologies in the biomolecular and pharmaceutical fields, such as high throughput screenings (HTS) and omics, have prompted the role for AI in drug discovery. Machine learning (ML) is a subtype of AI which allows the building of software algorithms capable of learning from massive datasets [15], such as the ones used for drug virtual screening. Based on the input supplied, ML algorithms can make predictions on new datasets by the implicit learning from the initial training dataset used for its construction.

Drug repositioning is not a new strategy but its application in drug discovery in conjunction with AI and computational tools is far from being a well-established, standard practice [16]. Some recent examples of drug repositioning efforts for RDs, employing AI and more specifically ML, are mentioned below.

Zhu, L. et al., used a hybrid computational drug repositioning approach, employing both transcriptomics and molecular docking methods, to successfully identify potential drug candidates for the rare Noonan and LEOPARD syndromes [17]. Ekins, S. et al., performed a HTS of approved drugs to identify potential ion channel inhibitors to treat Pitt Hopkins syndrome. Validated Bayesian ML models were used to predict the activity of these compounds based on their structure (structure-activity relationship models) [18]. To identify possible drugs for Huntington’s disease, Battista, T. et al., performed a virtual screening to predict the interaction ability of FDA-approved drugs with the sigma-1 receptor protein (σ1R), followed by molecular docking analysis of top candidates and evaluation in patient-derived fibroblast cell lines [19]. Selected drugs proved able to directly bind σ1R in vitro and yield a positive therapeutic response. Lee, Y. et al., developed the URSAHD (Unveiling RNA Sample Annotation for Human Diseases) that uses ML techniques to identify molecular and mechanistic features in diseases. This method has been successfully used to predict effective chemotherapeutics such as cisplatin used against refractory anemia with excess blasts (RAEB), and iron chelators such as resveratrol for sideroblastic anemia (SA) [20]. Similar drugs have previously demonstrated therapeutic efficacy for RAEB and SA in clinical studies, validating this approach [21,22].

Computational approaches have been employed not only to identify repositioning candidates, but also to systematically identify relevant drug targets for complex and multisystem diseases. Therefore, mapping key biological targets and cellular pathways, with a special focus on these diseases allowed to define this biological activity space [23,24].

Congenital disorders of glycosylation (CDG) are a group of rare genetic disorders caused by defects in the synthesis and attachment of glycans to proteins and lipids, and the synthesis of glycosylphosphatidylinositol (GPI) anchors. The mutated genes and proteins involved in some types of CDG are shown in Table 1. CDG are typically multi-systemic diseases with neurological manifestations observed in most patients [25]. More than 160 CDG types have been reported which are associated with approximately 210 phenotypes [26]. PMM2-CDG, the most common CDG, has a prevalence higher than 1/20,000 [27]. CDG may be broadly classified into four categories: (I) N-glycosylation defects; (II) O-glycosylation defects; (III) GPI-anchor biosynthesis and lipid glycosylation defects; and (IV) multiple and other glycosylation pathways defects [28].

Around 2500 patients have been diagnosed with CDG in Europe [29]. There is still no cure for CDG although a few CDG types are treatable with nutritional interventions [30]. The development of effective therapies for CDG is complicated by the broad genetic and clinical heterogeneity of this group of diseases. Not only is there a phenotypic heterogeneity between the various types, but also within the same CDG type (e.g., depending on the genetic variant). Phenotypic variability has been reported in patients carrying the same genetic variants [31,32], and possible involvement of genetic modifiers has been envisaged [33,34,35]. The last few years have seen significant advancements in CDG therapy development with a few promising treatments currently under pre-clinical or clinical evaluation. The therapies include substrate or co-factor supplementation, proteostasis inhibitors, pharmacological chaperones, and antisense and gene therapy [36]. CDG drug development could be pushed by drug repositioning alone or AI-assisted approaches, as in the case of other RDs [14]. The identification of biomarkers and disease models is crucial to find a cure or novel treatments [28]. Therefore, drug repositioning for CDG will greatly benefit from reliable therapeutic biomarkers and disease models.

In this paper we present the results of a literature search on CDG therapies proposed after October 2017, with emphasis on drugs being repositioned and undergoing pre-clinical and/or clinical trials, such as celastrol (pre-clinical), acetazolamide (phase 2) and epalrestat (phase 3) for PMM2-CDG and palovarotene (pre-clinical) for EXT1/EXT2-CDG. The chemical structures of these drugs are shown in Figure 1. We also summarize the latest biomarkers and disease models for CDG as well as the results of a recent survey addressed to the CDG community to determine key factors hampering therapy development.

## 2. Methods

### 2.1. Literature Analysis for Drug Repositioning in CDG

The literature was reviewed following the Preferred Reporting Items for Systematic Reviews and Meta-Analyses (PRISMA) guidelines. A search on the Medline database using PubMed as the search engine was conducted using a combination of 71 different keywords, targeting articles related to CDG, animal models, biomarkers and therapeutic approaches, particularly drug repositioning, released between October 2017 and March 2022. The selected keywords are listed in the Appendix A.

The inclusion/exclusion criteria were the following:(a)Only English-written manuscripts were included;(b)Articles reporting biomarkers, in vitro and/or in vivo models, compassionate use or clinical trials of therapies in CDG related to drug repositioning, were included;(c)Only articles reporting CDG with therapies related to drug repositioning, under development (compassionate use, clinical research) or already approved therapies were included;(d)Reviews were excluded, although we have included some for contextualization purposes.

For information about clinical trials, both the European and American web pages were consulted (EU Clinical Trials Register—Update, Home—ClinicalTrials.gov, accessed on 24 March 2022).

### 2.2. Stakeholders’ Views on AI for Drug Development in CDG

A lack of awareness and available information are amongst the most common impediments to drug research and development (R&D) in RDs. The “Assessing CDG needs and solutions for future therapies” survey was launched shortly before the 5th World Conference on CDG in 2021 to perform a comprehensive assessment of the awareness level and knowledge of stakeholders in the CDG community (CDG professionals, patients and laymen) [37]. A section of the questionnaire focused on stakeholder perspectives on drug profiling and AI for drug R&D. The results were examined and presented in this work.

## 3. Results

### 3.1. Literature Analysis

A total of 460 articles were found. The duplicates (138) were excluded and 322 articles were selected for further analysis. Due to the high number of results, keyword refinement was applied, leading to the exclusion of articles that were out of the scope of this work. 215 articles were included for title and abstract selection and a total of 42 articles meeting the inclusion criteria were selected for full paper analysis. The PRISMA flow diagram in Figure 2 summarises the screening process.

### 3.2. Disease Models

Modelling RDs by genetic modification in cellular or animal models is a valuable tool to efficiently recapitulate disease phenotypes and/or pathophysiological mechanisms allowing to test a large number of new or repositioned chemical compounds in a short period of time. In vitro models (i.e., cell lines such as CHO and HeLa), have been used to identify the molecular mechanisms involved in CDG, categorize gene variants as disease-causing and investigate protein functionality. Patient-derived cells, such as fibroblasts, have also been extensively used to study disease mechanisms and potential therapeutic strategies [28]. Recently, patient-derived lymphoblastoid B cell lines have been proposed as a model for studying CDG [39].

Lao et al., developed yeast models of PMM2-CDG that allowed for the analysis of evolutionarily conserved genotype-phenotype relationships across yeast and PMM2-CDG patients. In this model, growth is deficient in PMM2 mutants and correlates with the residual enzymatic activity. These patient avatars were used in a growth-based phenotypic drug repositioning screen of a library of 2560 approved and experimental drugs, compounds and natural products [40]. These yeast models are also attractive to identify molecular mechanisms of genetic compensation in PMM2-CDG. Experimental evolution has recently revealed compensatory mutations that restore growth and protein glycosylation in PMM2-CDG yeast. Some mechanisms have already been proposed to explain this correlation. Most importantly, this work opens the way for new insights for unraveling the molecular basis of PMM2-CDG [41].

Nevertheless, some limitations can only be overcome by using more complex organisms. To gain full knowledge of the disease pathophysiological mechanisms as well as to gather pre-clinical research data concerning absorption, distribution, metabolism, excretion and toxicity (ADME-Tox) parameters, essential for further clinical research, multicellular model organisms are necessary. Encountered difficulties in CDG models are particularly related to embryonic and neonatal lethality and a lack of replication of the patients’ disease phenotype. New techniques such as clustered regularly interspaced short palindromic Repeats (CRISPR)/Cas9 or conditional knockouts [42] have made in vivo gene engineering more accessible and efficient. In the past five years, several in vivo models have been developed for CDG, allowing the identification of pathophysiologic mechanisms [43,44,45] and the pre-clinical testing of drug repositioning candidates [46,47]. A compilation of both in vitro and in vivo disease models reported since October 2017 is presented in Table 2. In addition, we are aware of two novel *C. elegans* SRD5A3 knockout strains that have been generated for the screening of repositioned drugs for SRD5A3-CDG [48].

### 3.3. Biomarkers

Transferrin, an abundant serum glycoprotein, is the primary biomarker for CDG screening and therapy monitoring. Nevertheless, the existence of CDG with normal transferrin glycosylation, the normalization of glycosylation patterns with age in several CDG patients and charge-altering transferrin variants, stress the need for complementary biomarkers. Electrospray ionization mass spectrometry (ESI-MS) has been used to complement transferrin analysis [72]. *N*-glycome profiling using matrix-assisted laser desorption ionization time of flight (MALDI-TOF) MS broadens the scope of this technique by analyzing the entire *N*-glycans in a sample. It is increasingly used in CDG diagnosis, particularly when standard transferrin analysis does not reveal abnormalities [72,73]. Chen et al., described a new MS-based approach for CDG diagnosis, RapiFluor MS, that paves the way to the integration of *N*-glycomics approaches for diagnosis of glycosylation disorders [74]. In a recent glycomics study, specific glycomarkers for congenital disorders of N-glycosylation (CDG type I) were identified, which could be potentially used for diagnosis and therapy recording [75].

There are still few biomarkers available to monitor disease progression or therapy assessment. Recently, sorbitol has emerged as a biomarker for PMM2-CDG as urinary sorbitol levels seem to correlate with disease severity in patients. It was also used to monitor the effect of epalrestat (a repurposed drug in trial for PMM2-CDG) treatment in a pediatric PMM2-CDG patient [76].

Other improvements concern known therapies. An example is the PGM1-CDG Treatment Monitoring Index (PGM1-TMI), proposed to evaluate the efficacy of D-galactose supplementation in PGM1-CDG patients. The index allows one to track the glycosylation profile of transferrin during D-galactose supplementation and dose adjustment if necessary [77].

### 3.4. Drug Repositioning

#### 3.4.1. Celastrol for PMM2-CDG

Celastrol has been studied primarily for its anti-inflammatory properties as a drug for cancer, inflammatory and autoimmune diseases [78]. This molecule also regulates other cellular mechanisms (such as proteostasis) [79].

Protein homeostasis, or proteostasis, involves a set of cellular functions that ensures a functional proteome within the cell. To be functional, proteins need to acquire and maintain a proper quaternary structure and localization. The proteostasis network, consisting of molecular chaperones, the autophagy and proteosome systems, not only are responsible to assist proteins during the folding process as well as act as a quality control mechanism [80]. Proteostasis regulators have been proposed as therapeutics for rare diseases [81,82], including conformational diseases, such as PMM2-CDG [83,84]. As such, celastrol, which modulates the proteostasis network by activating the heat shock response (HSR) (Figure 3), has been successfully tested in patient-derived fibroblasts carrying several pathogenic PMM2 variants.

Celastrol treatment increased PMM2 protein levels and enzymatic activity in the disease cell models overexpressing genetic variants affecting PMM2 protein folding (namely D65Y, R162W, T237M) and PMM2 dimerization (F119L). Furthermore, several molecular chaperones from the HSR family were increased, both at the transcriptional and proteome levels [83,85]. These results represent the proof-of-concept for the use of stabilizing molecules as therapy for PMM2-CDG [85].

#### 3.4.2. Acetazolamide for PMM2-CDG

Nearly a thousand people have been reported with PMM2-CDG [86]. Most of them present with cerebellar syndrome and stroke-like episodes (SLE). The AZATAX trial (EudraCT number 2017-000810-44) was conducted to evaluate the efficacy and safety of acetazolamide in the treatment of cerebellar syndrome in PMM2-CDG patients. The rational for clinical evaluation of acetazolamide was based on the hypothesis that hypoglycosylation of calcium channels is an underlying pathomechanism of ataxia and SLE in PMM2-CDG patients [87]. This defect in glycosylation could cause an undesirable increase in calcium, a known inhibitor of PMM2 [88]. Acetazolamide (Figure 3), a non-competitive inhibitor of carbonic anhydrase, can act on the transmembrane potential by interfering with pH [89].

Acetazolamide is an approved drug that is used to treat retinal complications, epilepsy and other diseases, particularly in children. To date, no interference with other therapies has been reported [90,91,92].

In the AZATAX trial, twenty-four PMM2-CDG patients (mean age of 12.3 ± 4.5 years) were included in the European phase 2 clinical trial. The trial comprised a six-month first-phase single acetazolamide therapy, followed by a randomized five-week withdrawal phase. No serious side effects were reported, despite the requirement of dose adjustment in thirteen patients due to low bicarbonate levels or asthenia. Improvements on the International Cooperative Ataxia Rating Scale (ICARS), the Nijmegen Pediatric CDG Rating Scale (NPCRS) and a syllable repetition test (PATA test) were observed in 18 patients (75%) after 6 weeks of treatment. Despite clinical improvement, no relevant benefit on quality of life (QoL), apart from the anxiety score, was observed. Improvement of coagulation factors (prothrombin time, factor X, and antithrombin) was also observed. The drug was well tolerated in most patients and improved motor and cognitive features of the cerebellar syndrome [93].

Another clinical trial to assess the effect of acetazolamide on improving ataxia and to evaluate adverse events related to a longer administration is being conducted (ClinicalTrials.gov NCT04679389, accessed on 24 March 2022).

#### 3.4.3. Epalrestat for PMM2-CDG

A pilot drug repositioning study using a novel yeast PMM2-CDG model resulted in the identification of three compounds that restore the growth defect in mutant strains. One of the compounds is α-cyano-4-hydroxycinnamic acid, a potent aldose reductase inhibitor [40] which paved the way for the work of Iyer et al. Among 20 drugs, epalrestat was identified as a potential hit. Epalrestat (Figure 3), also an aldose reductase inhibitor, was shown to increase PMM2 activity in four PMM2-CDG patient fibroblast lines with genotypes R141H/F119L, R141H/E139K, R141H/N216I and R141H/F183S [47].

Epalrestat is already used to treat neuropathy as a diabetic complication [94]. The drug is widely marketed in Asia but has not yet been approved in the US or the EU.

Considering that aldose reductase catalyzes the conversion of glucose to sorbitol, its inhibition should increase the glucose availability for the synthesis of α-D-glucose-1,6-bisphosphate, a known PMM2 activator [95]. A single-patient phase 1 clinical trial has been ongoing since January 2020 to evaluate the tolerability of oral epalrestat monotherapy in a child with PMM2-CDG. The patient showed improvement of appetite, of body mass index (BMI) and of serum transferrin isoelectric focusing. There was minimal improvement of the NPCRS and of the ICARS ataxia score. No adverse events were observed during the trial period [76]. A phase 3, prospective, single-center clinical trial designed to assess the safety, tolerability, metabolic improvement and probable benefit of oral epalrestat therapy in pediatric subjects with PMM2-CDG will be conducted (ClinicalTrials.gov NCT04925960, accessed on 24 March 2022).

#### 3.4.4. Palovarotene for EXT1/EXT2-CDG

EXT1 and EXT2 are tumor suppressor genes that encode glycosyltransferases involved in the biosynthesis of heparan sulfate. Genetic variants of these genes cause multiple exostoses, osteochondromatosis, or EXT1/EXT2-CDG, which are autosomal dominant *O*-linked glycosylation disorders characterized by the formation of multiple cartilage-capped tumors (osteochondromas) [96].

Impaired heparan sulfate biosynthesis causes an increase in bone morphogenetic protein (BMP) signaling, which most likely leads to the formation of osteochondromas [42,97]. Palovarotene (Figure 4), a retinoic acid receptor and selective agonist for EXT1/EXT2-CDG, has already shown clinical safety for fibrodysplasia ossificans progressiva and emphysema [98,99]. The rational for testing palovarotene for the treatment of EXT1/EXT2-CDG is based on its ability to reduce heterotopic ossification in mouse models (likely by the inhibition of BMP signaling).

A pre-clinical trial was performed in a Fsp1^Cre^; Ext1^flox/flox^ mouse model of multiple osteochondromatosis. Four-week daily treatment with palovarotene, starting at postnatal day 14, reduced the number of developing osteochondromas by up to 91% in a dose-dependent manner [42]. It was also shown that the drug restores proper chondrocyte differentiation in Ext1-deficient progenitor cells in vitro. In addition, insights were provided concerning the relationship between palovarotene and BMP signaling [46]. These encouraging results pave the way for the first clinical trial for EXT1/EXT2-CDG.

A randomized, double-blind, placebo-controlled phase 2 clinical trial (ClinicalTrials.gov NCT03442985, accessed on 24 March 2022) was conducted between 2018 and 2020. The trial was terminated early to analyze the accumulated data and evaluate the efficacy and safety of palovarotene in osteochondromatosis. After the evaluation of these results, Ipsen (the trial sponsor) was granted authorization to initiate a new trial to evaluate the effect of palovarotene in patients with multiple osteochondromas over 14 years of age (ClinicalTrials.gov NCT05027802, accessed on 24 March 2022).

### 3.5. Stakeholders’ Views on AI for Drug Development in CDG

Prior to the 5th World Conference on CDG in 2021, a questionnaire was addressed to the CDG community to gather stakeholder’s perspectives on therapy development. The views of both CDG professionals (mainly researchers and, to a lesser extent, clinicians) and CDG families on the entire drug development process, from drug discovery to drug approval, were collected.

A particularly relevant outcome for the present work concerns the stakeholder’s opinion on AI tools. Most questionnaire participants believe in the potential of AI to accelerate the discovery of new therapies. In particular, professionals believe that AI potentiates the finding of novel chemical compounds (65.2%). However, only 23.9% of researchers reported to benefit from AI tools to date. Similarly, most families recognize AI as a method for disease model development (57.1%). Regardless of the final application, the majority of surveyed stakeholders believes that the power of AI lies in the ability to access various data resources simultaneously [37].

## 4. Discussion

Traditionally, the entire drug development process, from drug discovery to clinical approval, is a lengthy process spanning a decade or longer [100]. The financing efforts required for drug development [101] may be even greater for RDs [102], which are typically underrepresented in therapy development due to a combination of factors such as underdiagnosis and underestimation of disease frequency insufficient data on disease patterns, epidemiology, pathophysiology, biomarkers; patient outcomes; and low commercial incentives from pharmaceutical companies paired with lack of funding [14].

Drug repositioning is presented as an alternative to conventional R&D “from scratch” approaches, promising to overcome some of its obstacles, particularly for the underrepresented RDs. By redeveloping approved drugs for novel applications and introducing them in a new pathophysiological setting, it is possible to screen known compounds for their activity towards a different biochemical target in search for new therapeutic uses [9]. Drug repositioning allows researchers to potentially bypass assays and clinical trials, mostly regarding toxicological, pharmacodynamic, and pharmacokinetic concerns (as these have been previously addressed in the primary clinical studies) [103]. Although drug repositioning is an attractive strategy for drug developers (both scientists and investors), it is not yet sufficiently supported from the financial and legal perspective. Regulatory incentives and patent protection are the key points that would support investment in repositioned drug development [10].

ML technology is a great opportunity for drug development as, by predicting biochemical properties of compounds and target interactions, it is possible to filter molecules with desired features while discarding others unlikely to be effective from further investigational efforts. This strategy reduces the weight of in vitro assays, animal testing and clinical trials [12], which is especially advantageous for RDs where the lack of work models and patient trials is a limiting factor [104,105]. A noteworthy aspect is that the performance of ML, and AI in general, intrinsically depends on the quality and completeness of the datasets used. Thus, the development of increasingly accurate ML technology must be accompanied by progress in screening methodologies, to efficiently measure relevant information, such as chemical reactivity, biological activity, pharmacological endpoints and cell toxicity to enhance AI development in drug discovery [106]. Nowadays, the chemical-response data obtained from HTS is growing exponentially and contributes to the current establishment of big data approaches. The development of deep learning neural networks with the corresponding increase in chemical and biological data has resulted in a paradigm shift in data mining pertaining to the chemical-biological spaces [107]. Furthermore, some characteristics of RDs, starting from the various nomenclatures and classifications available, make it hard to integrate the available information. Recently, it has been highlighted that drug repositioning for RDs benefits from the creation of a systematic ontology to extract information from clinical trials and omics data [108].

Even though AI tools are still sporadically employed for CDG drug development, they have been successfully used for the diagnosis and characterization of these diseases [14]. Few advances in CDG therapy have been made over recent years [36]. By reviewing the literature, it is noted that four of the drugs advanced to different trial phases are repositioned drugs. They represent different possibilities for identifying repurposed compounds, namely through expert awareness on clinical manifestations, as in the case of acetazolamide (Figure 1A) and palovarotene (Figure 1C), by targeting specific pathophysiological mechanisms, as for celastrol (Figure 1D) or by screening libraries of chemicals, as for epalrestat (Figure 1B).

Based on previous observations, it was hypothesized that defects in the CaV2.1 channel are involved in cerebellar syndrome and SLE in PMM2-CDG patients. Acetazolamide could prevent not only these neurological complications but also ataxia, another major symptom of the disease [93]. Thus, research and clinical observations have allowed this drug, already in use for other diseases, to be investigated for PMM2-CDG in two separate clinical trials. Successful treatment of these symptoms would allow a significant improvement in the QoL of these patients.

Besides symptom management, an even more intricate issue concerns finding a cure for the disease. The search for a potential drug amongst approved molecules is favoured if the pharmacological target is fully explored. It follows that for RDs, particularly lacking in data, this knowledge-based approach is often not applicable. AI tools hold a great potential for accelerating therapy options (e.g., by allowing one to match thousands of molecules with any pharmacological target, thus playing an increasingly important role in drug repositioning for RDs, including CDG) [14,109]. Epalrestat is a successful example of organic intelligence application for drug repositioning in CDG. The drug emerged from the HTS of repositioned compounds, and thus with a systematic and data-oriented approach. The promising results in yeast and patient fibroblast models of PMM2-CDG (i.e., increasing enzyme activity of some variants and restoring disease models’ phenotype) suggest that epalrestat may be the first drug capable of rescuing PMM2 activity, and not only on some symptoms of the disease.

Although still in a preliminary experimental stage, another repositioned drug for PMM2-CDG is being tested. Administration of celastrol, a proteostasis regulator, resulted in an increase in PMM2 enzyme activity in patient-derived cell lines. The drug is being examined since this type of molecules has proved effective for several rare diseases characterized by protein unfolding, such as PMM2-CDG [83].

Once repositioning hits are identified in silico and/or in vitro, their efficacy on the new target must be verified using disease models. Findings from a pre-clinical study in a mouse model of EXT1/EXT2-CDG proved the effectiveness of palovarotene in reducing the development of benign bone tumors, suggesting it could be repurposed to treat this CDG [46]. As mentioned above, this is a typical drug repositioning strategy: the investigation arises from insights based on the observation of typical clinical presentations. In this case, palovarotene is directed at benign bone tumors that characterize EXT1/EXT2-CDG, and a trial is being conducted in mice modeling this phenotypic feature.

Given the potential of drug repositioning for CDG, it is not surprising that more efforts in this area are under development. The pharmaceutical company Modelis has developed an in-house pipeline for drug repositioning. In the first stage, phenotype-based drug screening of a 4500-compound library is carried out in worm disease models to identify candidates for further investigation. This pipeline is being used to discover therapeutic candidates for SRD5A3-CDG. To date, several hits have been identified and are undergoing validation [110].

Several CDG models are already available and efforts to identify therapeutic biomarkers and tractable animal models are ongoing. A recent survey addressed to the CDG community stakeholders revealed that the accessibility of biobanks and disease models is limited [37]. Given the fundamental importance of these tools in the drug development process, especially in the early (pre-clinical) phases, disseminating and sharing existing disease models is crucial to foster advances in CDG therapies.

## 5. Conclusions

The present work outlines two areas of opportunity to accelerate drug development for RDs. Drug development, especially for RDs, is aided by advances in AI and by the availability of good models. Here we review the progress in these fields since our 2018 publication [28], collecting the latest therapeutic biomarkers and disease models for CDG. The other aspect is drug repurposing, illustrated by four repositioned drugs undergoing trials as promising therapies for two CDG types.

## Figures and Tables

**Figure 1 ijms-23-08725-f001:**
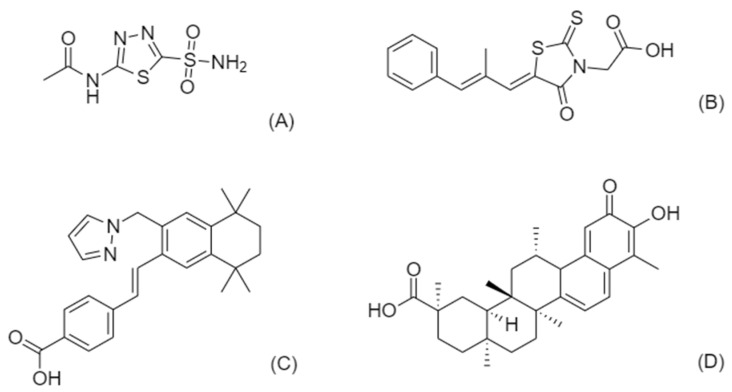
Chemical structures of CDG-repositioned drugs in clinical pipelines. (**A**) acetazolamide, (**B**) epalrestat, (**C**) palovarotene, and (**D**) celastrol.

**Figure 2 ijms-23-08725-f002:**
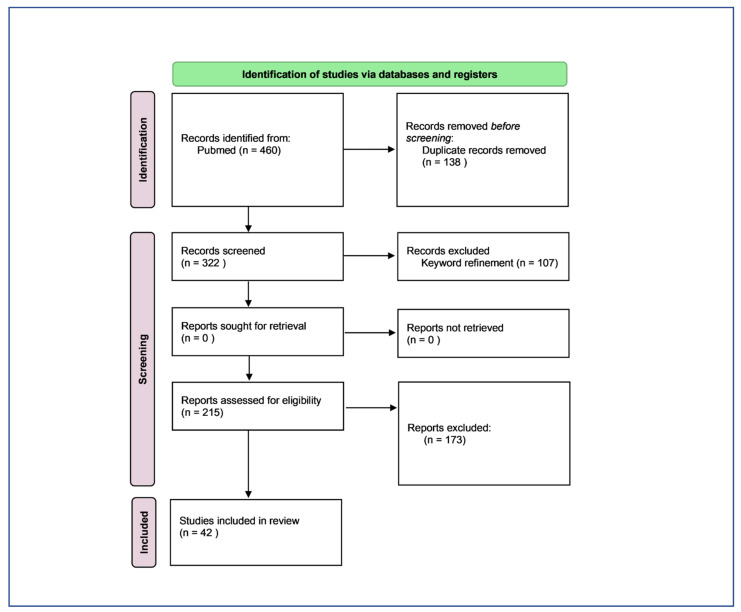
PRISMA flow diagram adapted from the study by Page et al. [38].

**Figure 3 ijms-23-08725-f003:**
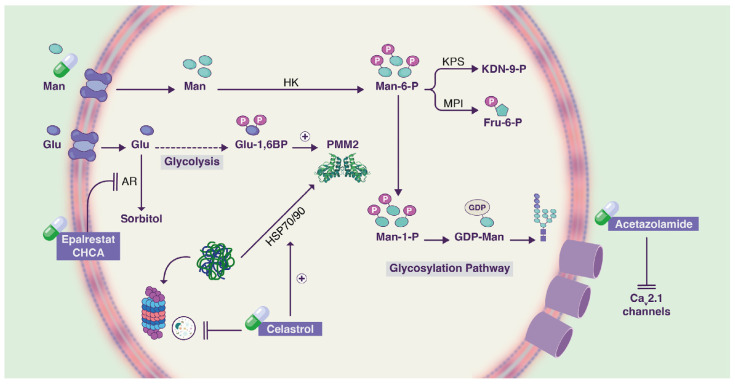
Schematic representation of the mechanism of action of drugs under testing in clinical trials for PMM2-CDG. Acetazolamide inhibits CaV2.1 calcium channels. Epalrestat inhibits sorbitol synthesis. Celastrol modulates PMM2 proteostasis. Adapted from Gámez et al., 2020 [85].

**Figure 4 ijms-23-08725-f004:**
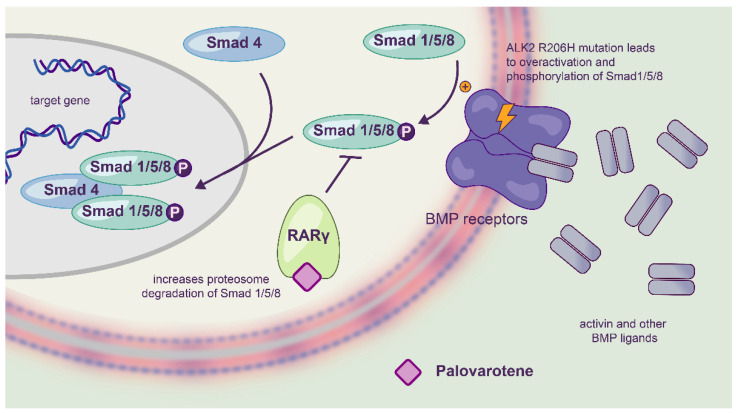
Schematic representation of the mechanism of action of palovarotene.

**Table 1 ijms-23-08725-t001:** Overview of mutated genes and respective proteins involved in some CDG. The gene identification number (ID) is taken from the Gene database of NCBI.

*Gene* (Gene ID)	Protein	Disorder
*ALG2* (85365)	alpha-1,3/1,6-mannosyltransferase	ALG2-CDG
*ALG13* (79868)	UDP-N-acetylglucosaminyltransferase (subunit)	ALG13-CDG
*B4GALT1* (2683)	beta-1,4-galactosyltransferase 1	B4GALT1-CDG
*CAD* (790)	carbamoyl-phosphate synthetase 2, aspartate transcarbamylase, and dihydroorotase (enzyme complex)	CAD-CDG
*COG4* (25839)	component of oligomeric golgi complex 4	COG4-CDG
*COG5* (10466)	component of oligomeric golgi complex 5	COG5-CDG
*COG7* (91949)	component of oligomeric golgi complex 7	COG7-CDG
*DPAGT1* (1798)	dolichyl-phosphate N-acetylglucosaminephosphotransferase 1	DPAGT1-CDG
*EXT1* (2131)	exostosin glycosyltransferase 1	EXT1-CDG
*EXT2* (2132)	exostosin glycosyltransferase 2	EXT2-CDG
*FUT8* (2530)	fucosyltransferase 8	FUT8-CDG
*GMPPB* (29925)	GDP-mannose pyrophosphorylase B	GMPPB-CDG
*GNE* (10020)	glucosamine (UDP-N-acetyl)-2-epimerase/N-acetylmannosamine kinase	GNE-CDG
*MAGT1* (84061)	magnesium transporter 1	MAGT1-CDG
*MOGS* (7841)	mannosyl-oligosaccharide glucosidase	MOGS-CDG
*MPI* (4351)	mannose phosphate isomerase	MPI-CDG
*NANS* (54187)	N-acetylneuraminate synthase	NANS-CDG
*PGM1* (5236)	phosphoglucomutase 1	PGM1-CDG
*PGM3* (5238)	phosphoglucomutase 3	PGM3-CDG
*PIGA* (5277)	phosphatidylinositol glycan anchor biosynthesis class A	PIGA-CDG
*PMM2* (5373)	phosphomannomutase 2	PMM2-CDG
*SLC35C1* (55343)	solute carrier family 35 member C1	SLC35C1-CDG
*SLC39A8* (64116)	solute carrier family 39 member 8	SLC39A8-CDG
*SRD5A3* (79644)	steroid 5 alpha-reductase 3	SRD5A3-CDG
*ST3GAL3* (6487)	ST3 beta-galactoside alpha-2,3-sialyltransferase 3	ST3GAL3-CDG
*ST3GAL4* (6484)	ST3 beta-galactoside alpha-2,3-sialyltransferase 4	ST3GAL4-CDG
*ST3GAL5* (8869)	ST3 beta-galactoside alpha-2,3-sialyltransferase 5	ST3GAL5-CDG

**Table 2 ijms-23-08725-t002:** Overview of in vitro and in vivo disease models for congenital disorders of glycosylation (CDG) reported since October 2017 to March 2022. NR: not reported; MO: morpholino oligonucleotide; KO: knockout; KD: knockdown.

Defect	CDG	Cell/Organism	Model	Major Findings/Phenotype	Reference
N-linked glycosylation	ALG2-CDG	*Oryzias latipes* (medaka)	*Alg2* ^+/p.G336^ *Alg2* ^p.G336/p.G336^	Modelling ALG2-CDG patient phenotypes, in terms of morphology (facial skeleton and neuronal defects) and hypo-N-glycosylation (especially affecting rod photoreceptors)	[49]
	ALG13-CDG	*Mus musculus* (mouse)	*Alg13* KO	- Increase of the severity of kainic acid (KA)-induced and pilocarpine-induced seizures - Exacerbation of the classical pathological manifestations of epilepsy in KA-induced epileptic mice	[44]
	DPAGT1-CDG	*Xenopus laevis*	*Dpagt1* KO (mRNA)	- Posteriorization of *X. laevies* embryos - Significantly altered expression of *Wnt* reporter genes, *Xnr3* and *chrd*	[45]
		*Danio rerio* (zebrafish)	*Dpagt1* KO (mRNA)	Inhibition of eye formation	
	FUT8-CDG	Mouse	*Fut8* ^−/−^	High mortality rate after birth due to respiratory defects and severe growth retardation	[50]
	MAGT1-CDG	Jurkat cell line	*Magt1* ^−/−^	Selective deficiency of *N*-glycoproteins and glycosylation defects in immune-response proteins such as CD28	[51]
		Human embryonic kidney (HEK) 293T cell line	*Magt1* KO *Magt1/Tusc3* KO	- MAGT1 and paralog protein TUSC3 are OST subunits and their role in glycosylation is interchangeable - MAGT1 and TUSC3 have different tissue distribution	
	MOGS-CDG	*Schizosaccharomyces pombe* (yeast)	Δ*gls1-S*	- Abrogated G3M9 deglucosylation - Lack of triglucosylated glycoprotein deglucosylation - Distortion of cell wall and absence of underlying ER membranes	[52]
	MPI-CDG	TWNT-4 and LX-2 ^a^ human hepatic stellate cells	*Mpi* KD (siRNA)	- ^a^ Depletion of MPI activity - ^a^ Increased expression of *COL1A1*, *PDGFRB*, and *ACTA2*.	[53]
	PMM2-CDG	*Caenorhabditis elegans*	*Pmm2* ^F125L/F125L^	- Larval lethality not seen, growth defects or any observable locomotor defects in liquid media - Reduced PMM enzyme activity - Sensitive to tunicamycin and bortezomib (induces larval arrest in worms)	[47]
		*Saccharomyces cerevisiae* (yeast)	*Sec53*Δ *Sec53*^E146K (E139K)^ *Sec53*^V238M (V231M)^ *Sec53*^F126L (F119L)^ *Sec53*^E100K (E93A)^ *Sec53*^R148H (R141H)^	Drug repurposing screen revealed three novel chemical modifiers that subdued growth defects in SEC53 protein variants	[40]
		Zebrafish	*Pmm2* KD (MO) *Mmp2* KD (MO) *Mmp9* KD (MO) *Furina* KD (MO)	Reducing proconvertase activity restores matrix metalloproteinase (mmp) activity and improves *N*-cadherin processing	[54]
		EBV-transformed lymphoblastoid B cell lines (B-LCL) from 13 patients		Carbonic anhydrase 2 is proposed as a cellular biomarker for CDG	[39]
O-linked glycosylation	B4GALT1-CDG	Mouse embryonic stem cells (mESCs)	*B4Galt1* KO	Enhanced resistance to ricin	[55]
	CRPP-CDG	Mouse	*FKRP* ^P448L/P448L^	- Early onset of dystrophic pathology - Undetectable levels of F-α-DG in cardiac and skeletal muscles	[56,57]
	EXT1/EXT2-CDG	Mouse	*Col2a1-Ext1^CKO^* stochastic KO	- Macroscopic osteochondromas development in bones by P28 - Abnormal cell clusters in Ranvier grooves	[42]
			*Fsp1-Ext1^CKO^* (perichondrium-targeted *Ext1*–conditional KO)	Development of multiple osteochondromas	
			*Ext1^f/f^ Agr-CreER*	- Osteochondroma formation 6 to 8 weeks of tamoxifen injection - Marked decrease in immunodetectable pERK1/2 levels	[58]
			*Ext1^f/f^ Col2-CreER*	- Cranial base defects - Disorganized synchondroses - Deranged growth plate-like organization - Osteochondromas development	
GPI-biosynthesis	PIGA-CDG	Human male colon cancer cell line (HCT116)	*Piga*Δ	NR	[59]
		Mouse	^a,b^ In-M-cko ^a,c^ Ex-M-cko Th-H-cko	- ^a^ Impaired long-term fear memory - ^a^ Increased susceptibility to KA-induced seizures - ^b^ Severe limb-clasping phenotype - ^c^ Changes in hippocampal synapses	[60]
Multiple and other glycosylation pathways	CAD-CDG/ Enzyme complex (ATase, CPSase, ATCase and DHOase)	Human U20S cells	*CAD* KO (homozygous c.70delG frameshift (p.Ala24-Profs*27) within exon 1 using CRISPR/Cas9)	No expression of CAD protein	[61]
	GMPPB-CDG	Zebrafish	*Gmppb* KD (MO)	Gmppb involvement in neuronal and muscle development	[62]
	GNE-CDG	Chinese hamster ovary (CHO) cell line	*Gne* KO	- CMP-sialic acid reduction - Decreased sialylation of cell surface glycans	[63]
	COG4-CDG	RPE1 and HEK293T cell lines	*Cog4* KO	- Expression of G516R and R729W rescues the COG4 KO phenotypes - COG4 G516R and R729W do not alter Golgi morphology - O-glycosylation defect in cells expressing COG4 G516R and N-glycosylation defect in cells expressing COG4 R729W mutants	[64]
	COG5-8	*S. cerevisiae*	*Cog5-8*Δ (*cog5-8::kanMX6*)	- Relocation of individual COG subunits to mitochondria - Recruitment of a limited number of other COG subunits to mitochondria	[65]
	COG5-CDG	*Drosophila melanogaster*	P element insertion mutations in the Cog5 (*fws*) subunit	Impairment of spermatocyte cytokinesis, acroblast structure and elongation and individualization of differentiating spermatids	[66]
	COG7-CDG	*D. melanogaster*	*Cog7* ^z4495/z5797^	- Altered *N*-glycan profile - Pronounced neuromotor defects - Reduced lifespan	[67]
	NANS-CDG	CHO cell line	*Nans* KO	CMP-sialic acid reduction	[63]
	PGM1-CDG	Mouse	*Pgm2* ^−/−^	Embryonic lethality	[43]
			*Pgm2* ^+/−^	- Profound decrease of the tetrasialotransferrin glycoform (type 1), and relative increase of truncated glycans (type 2 pattern) - No increase in mannosylation and fucosylation - A glycan-processing defect, but different from biallelic PGM1 mutant human cells	
	PGM3-CDG	*D. melanogaster*	*DPgm3* KO (RNAi)	- Notches at the adult wing margin - Severe reduction of *sens* expression along the entire dorsoventral boundary	[45]
		*Xenopus laevis*	*Pgm3* (mRNA)	Posteriorization of embryos	
			*Pgm3 KO* (MO)	Anteriorization of embryos	
		Zebrafish	*Pgm3* (mRNA)	Inhibition of eye formation	
	SLC35C1-CDG	mESCs (haploid state)	*Slc35c1* ^−/−^	Lack of fucosylated structures	[55]
		Mouse intestinal organoids		Improved ricin resistance	
	SLC39A8-CDG	Mouse	ZIP8-iKO (*Slc39a8^fl/fl^ UBC-CreERT2*)	- Reduced expression of *slc39a8* in liver, brain, kidney and small intestine - Reduced levels of Mn in whole blood and tissues - Defective protein N-glycosylation - Hypogalactosylation	[68]
			ZIP8-LSKO (*Slc39a8^fl/fl^ Alb-Cre*, a liver-specific KO)	- Decreased *Slc39a8* mRNA levels in liver - Decreased Mn levels in liver, kidney, brain and heart - Decreased whole blood Mn levels	
	SRD5A3-CDG	Mouse	Cerebellar conditional KO En1-Cre; *Srd5a3fl/^-^*	- Motor coordination defects - Abnormal granule cell development - Mild N-glycosylation impairment - Major ER homeostasis alteration	[69]
	ST3GAL3-CDG	Mouse	*St3gal3* KO	Minor hematologic abnormalities	[70]
			*St3gal2/st3gal3* double KO	Lack of GD1a and GT1b gangliosides	
	ST3GAL4-CDG	KBM7 ST3GAL4 KO-1 and KO-2 cells	*ST3GAL4* ^−/−^	- Loss of sialyl Lewis X - Increased sensitivity to ricin	[55]
	ST3GAL5-CDG	HEK 293T	G342S- C195S ^a^- G201A ^a^- E355K-HaloTag-ST3GAL5	^a^ Complete loss of GM3 synthase activity	[71]

## Data Availability

Not applicable.

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
