# Peer review of "Systematic Review: Drug Repositioning for Congenital Disorders of Glycosylation (CDG)"

_ijms, 2022, doi:10.3390/ijms23158725_

Round 1

Reviewer 1 Report

The paper is interesting and well written. The question is, whether it fits to this Journal. Let the Editor decide.

Author Response

Thank you for the positive comments to our manuscript. 

Reviewer 2 Report

The authors summarized very recent progress on artificial intelligence and its applications in drug repositioning for rare diseases, focusing on the congenital disorders of glycosylation. The work is of great importance.  The review has been well-organized and presented. There are  appropriate and adequate references to support the conclusion. I am satisfied with the current format and have no further comments or correction on the manuscript.

Author Response

(The authors gave the same response as above.)

Reviewer 3 Report

The article Systematic Review: Drug repositioning for Congenital Disorders of Glycosylation (CDG) is good for publishing. However, before accepting, some revisions are required.

1)      Provide a table in introduction listing the names of important  congenital disorders of glycosylation along with their respected proteins

2)      Authors can merge this information in table 2 also creating new columns

3)      Drug repurposing strategies for CDGs may be described in more details

Author Response

Revisor 3 asked us to:

1-2)     Provide a table in introduction listing the names of important congenital disorders of glycosylation along with their respected proteins or merge this information in table 2 creating new columns.

We preferred the first option to ease the manuscript readility. Therefore, we have added table 1 in the introduction section and made minor changes to the table in the results section (table 2 in the revised version).

3)      Drug repurposing strategies for CDGs may be described in more details.

We truly appreciated the possibility to revise this part and we have added some details in the discussion section. New parts introduced for the revised version have been highlighted in green while sentences to be deleted are highlighted in red.

In addition, we have made minor changes to adjust the bibliography numbering.

Thank you for the positive comments to our manuscript. We are looking forward to hearing from you.